# NeuMA: Neural Material Adaptor for Visual Grounding of Intrinsic Dynamics

**Junyi Cao**[1][*]  **Shanyan Guan**[2][†]  **Yanhao Ge**[2]  **Wei Li**[2]  **Xiaokang Yang**[1]  **Chao Ma**[1][‡]

[1] MoE Key Lab of Artificial Intelligence, AI Institute, Shanghai Jiao Tong University
[2] vivo Mobile Communication Co., Ltd.

{junyicao, xkyang, chaoma}@sjtu.edu.cn
{guanshanyan, halege, liwei.yxgh}@vivo.com

## Abstract

While humans effortlessly discern intrinsic dynamics and adapt to new scenarios, modern AI systems often struggle. Current methods for visual grounding of dynamics either use pure neural-network-based simulators (black box), which may violate physical laws, or traditional physical simulators (white box), which rely on expert-defined equations that may not fully capture actual dynamics. We propose the Neural Material Adaptor (*NeuMA*), which integrates existing physical laws with learned corrections, facilitating accurate learning of actual dynamics while maintaining the generalizability and interpretability of physical priors. Additionally, we propose Particle-GS, a particle-driven 3D Gaussian Splatting variant that bridges simulation and observed images, allowing back-propagate image gradients to optimize the simulator. Comprehensive experiments on various dynamics in terms of grounded particle accuracy, dynamic rendering quality, and generalization ability demonstrate that NeuMA can accurately capture intrinsic dynamics. Project Page: https://xjay18.github.io/projects/neuma.html.

## 1 Introduction

Teaching a machine to "see, understand, and reason" the physical world like humans has been a fundamental pursuit of machine learning and cognitive science. Imagine the flexibility, robustness, and generalizability of human intelligence: simply by observing an object falling on the ground and bouncing up, even a young child can make a plausible guess of its intrinsic dynamics (by telling what material it is made of), adapt to new scenarios (involving new objects and initial conditions) and predict interactions with other objects. However, modern AI systems still fail to match this cognitive ability, known as visual grounding, of humans.

Many efforts have thus been made to impart the ability of visual grounding of dynamics to AI systems [5, 25, 26, 28, 45], typically by training a differentiable simulator [14, 31, 34, 72, 90] with pixel supervision from a differentiable renderer [37, 58]. Depending on the formulation of the simulator, current works can be categorized into two types: black box or white box. Black box approaches [26, 89] directly use a neural network to model the dynamic transition. Due to the deep coupling among extrinsic attributes (*e.g.*, geometry) and intrinsic (physical) motion during rendering, black box approaches are prone to violating physical laws and have limited generalization capability without any physics constraints on the transition process [25, 53].

In contrast, white box methods [45, 60, 95] use traditional physical simulators (*e.g.*, Material Point Method [34]) to approximate object dynamics explicitly with partial-differential equations (PDE), *a.k.a.*, motion equations. These methods back-propagate pixel differences from a renderer to the

---

[*]This work was done during an internship at vivo.   [†]Project lead.   [‡]Corresponding author.

38th Conference on Neural Information Processing Systems (NeurIPS 2024).

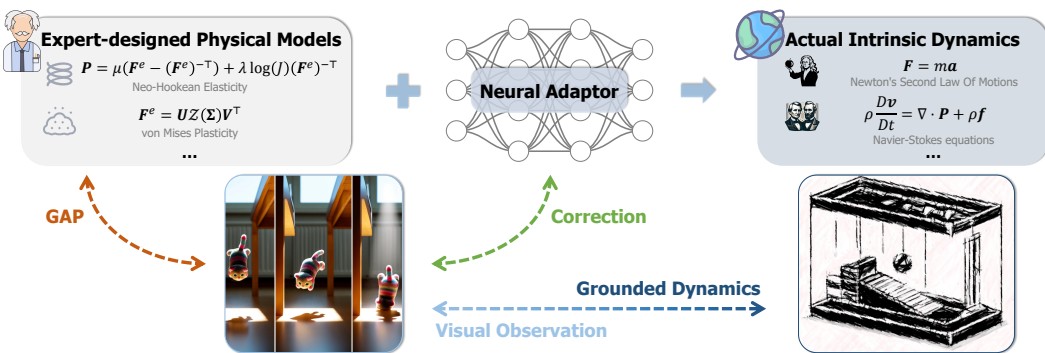

Figure 1: **The core idea of NeuMA**: Learning to correct existing expert knowledge on object motions by fitting a neural material adaptor to ground-truth visual observations.

physical coefficients (*e.g.*, Young's modulus, Poisson's ratio) of an analytical simulator. Naturally, the estimated physical coefficients can be applied to new scenes. However, the motion equations are expert-designed and may not perfectly align with the actual dynamics. This raises the key question we ask in this work: *how to accurately infer the actual intrinsic dynamics from the visual observations?*

To answer this question, we propose the Neural Material Adaptor (*NeuMA*), a learning-based model with physics-informed priors. As shown in Figure 1, the core idea of NeuMA is to formulate the learning of intrinsic dynamics specified by physical law $\mathcal{M}$ as a residual adaptation paradigm: $\mathcal{M} := \mathcal{M}_0 + \Delta\mathcal{M}$, where $\mathcal{M}_0$ is the expert-designed physical models, and $\Delta\mathcal{M}$ represents the correction term grounding to the observed images. This paradigm enjoys two advancements: on one hand, unlike white-box methods solely relying on $\mathcal{M}_0$, NeuMA can model the actual intrinsic dynamics by optimizing $\Delta\mathcal{M}$ to align with observations (*more accurate and flexible*); on the other hand, unlike black-box methods ignoring any physical priors, NeuMA fits the actual dynamics based on commonly-agreed physical models $\mathcal{M}_0$ (*more generalizable and physically interpretable*). Specifically, built upon the progress in physics simulation, NeuMA uses the Neural Constitutive Laws (NCLaw) [53] to formulate $\mathcal{M}_0$, which is a network that encodes existing physical priors and constraints. As for $\Delta\mathcal{M}$, we use a low-rank adaptor [30] that enjoys the efficient adaptation and preservation of the prior $\mathcal{M}_0$. Then, to supervise the simulation module via rendering, we propose Particle-GS, a differentiable renderer in the form of a particle-driven 3D Gaussian Splatting (3DGS) [37] variant. It leverages the predicted motion of particles to drive Gaussian kernels through a pre-defined relationship between particles and kernels. We use Particle-GS as the bridge from simulation to visual images, which allows marrying image gradients to optimize the simulator.

We evaluate NeuMA on various dynamic scenes with different materials and initial conditions. It shows competitive results in object dynamics grounding and dynamic scene rendering while achieving good generalization to novel shapes, multi-object interactions, and extended-time prediction.

## 2 Problem Formulation

We tackle the problem of grounding the intrinsic dynamics of an object from a sequence of visual observation $I = \{\boldsymbol{I}_1, \boldsymbol{I}_2, \ldots, \boldsymbol{I}_T\}$. Following common practice [18, 53, 60, 97], in this work, we adopt the elastodynamic equation [22] to describe the dynamical systems:

$$\rho\ddot{\boldsymbol{\phi}} = \nabla \cdot \boldsymbol{P} + \rho\boldsymbol{b}, \tag{1}$$

where $\nabla \cdot \boldsymbol{P}$ is the divergence of the stress tensor $\boldsymbol{P}$, $\rho$ is the object density, and $\boldsymbol{b}$ is the given body force. Besides, $\phi$ denotes the displacement field to describe the object's deformation, and $\ddot{\phi}$ is its acceleration. To solve Equation (1), material models $\mathcal{M}$ (see Appendix A for the background), which delineate how the object responds (depicted by $\boldsymbol{P}$) under deformations (depicted by $\phi$), should be prescribed. To realize differentiable grounding from observed videos, we then parameterize $\mathcal{M}$ with learnable parameters $\theta$ (*i.e.*, $\mathcal{M}_\theta$) [32, 53, 79]. In summary, the dynamical system governed by Equation (1) can be described by a transition model $\mathcal{S}$: $\boldsymbol{s}_{t+1} = \mathcal{S}(\boldsymbol{s}_t; \mathcal{M}_\theta)$, where $\boldsymbol{s}_t$ are the particle states (*e.g.*, positions and velocities) at the $t$-th time step.

To achieve dynamics grounding using only visual data, we develop a differentiable renderer $\mathcal{R}$. This enables pixel supervision to be used for backpropagation to the transition model $\mathcal{S}$, allowing for the

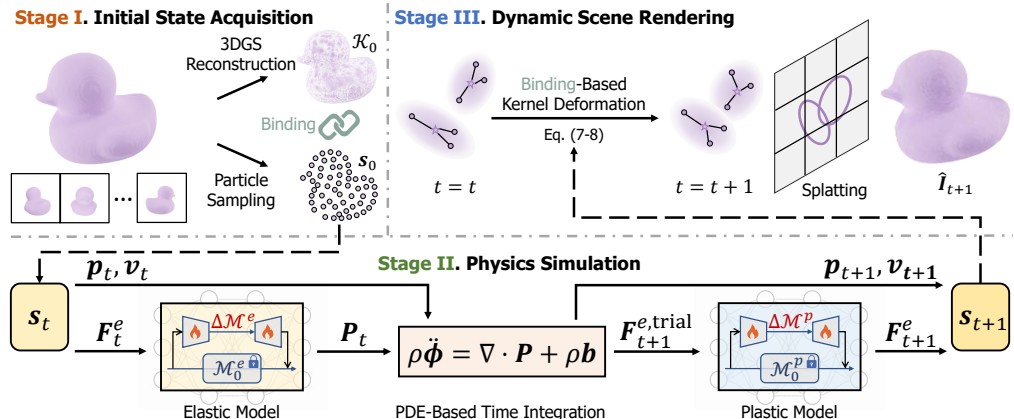

Figure 2: **The pipeline of NeuMA for visual grounding**. During Stage I, we first reconstruct the 3D Gaussian kernels of the foreground object using masked multi-view images. Then, we uniformly sample the initial physical particles from the object volume and bind them to the reconstructed Gaussian kernels. In Stage II, we integrate the neural material adaptor into the PDE-based simulation framework to estimate the actual dynamics. In Stage III, we deform the Gaussian kernels according to the binding relationship (pre-computed in Stage I) and then render 2D images. The neural material adaptor is trained end-to-end using the difference between the rendered and observed images.

optimization of the neural material model $\mathcal{M}_\theta$. Specifically, $\mathcal{R}$ learns to synthesize an image $\hat{I}_t$ given the state $s_t$, $i.e.$, $\hat{I}_t = \mathcal{R}(s_t; K_t, P_t)$, where $K_t, P_t$ denotes the camera's intrinsic and extrinsic matrix at the $t$-th time step. The optimal weights $\theta^*$ of $\mathcal{M}_\theta$ is obtained by minimizing the visual difference $\mathcal{L}_v$ between predicted images and the ground-truth observations:

$$\theta^* = \arg\min_\theta \mathcal{L}_v = \arg\min_\theta \|\hat{I}_t - I_t\|_2. \tag{2}$$

## 3 Method

We introduce NeuMA, a neural-network-based material adaptor, to learn the intrinsic dynamics specified by the material model $\mathcal{M}$ from only visual observations $I$. Our key insight is to implicitly represent the material model as a learnable, residual term $\Delta\mathcal{M}_\theta$ based on widely accepted physical priors $\mathcal{M}_0$ ($e.g.$, neo-Hookean elasticity for elastic objects [83], $etc$.). This gives:

$$\mathcal{M}_\theta \coloneqq \mathcal{M}_0 + \Delta\mathcal{M}_\theta, \tag{3}$$

where we implement $\Delta\mathcal{M}_\theta$ using the low-rank adaptation (LoRA) [30] to restrict our correction not to overturn the physical priors. We embed the whole material model $\mathcal{M}_\theta$ into a differentiable simulator $\mathcal{S}$, followed by a differentiable renderer $\mathcal{R}$ to enable supervised training directly on the outputs ($i.e.$, synthesized images) of $\mathcal{R}$. In view of recent advances in particle-based physical simulation, which achieve satisfactory performance for visual dynamics grounding [26, 45], we instantiate $\mathcal{S}$ as the well-known Material Point Method (MPM) [31, 34, 72] since it can handle complex motion simulation for various materials and easily back-propagate gradients for end-to-end training [26, 45, 85]. Please refer to Appendix E for more details about MPM. We further embrace the state-of-the-art scene representation technique, 3D Gaussian Splatting (3DGS) [37], to build up our differentiable renderer, Particle-GS, due to its explicit and flexible representation for efficient deformation. Please see Appendix A for details about 3DGS.

The overall pipeline for NeuMA to conduct visual dynamics grounding, as shown in Figure 2, includes three steps: initial state acquisition, physics simulation, and dynamic scene rendering.

### 3.1 Initial State Acquisition

Start with a set of calibrated multi-view observations $I_0 = \{I_0^1, I_0^2, \ldots, I_0^V\}$ depicting an object at the initial time step ($i.e.$, $t = 0$), we are interested in obtaining (1) the 3D Gaussian kernels $\mathcal{K}(i) = \{x(i), \alpha(i), A(i), c(i)\}$ representing the object, where $x(i), \alpha(i), A(i), c(i)$ denote the

center, opacity, covariance matrix, and spherical harmonic coefficients of the $i$-th Gaussian kernel; and (2) the initial state $s_0 = \{p_0, v_0, F_0^e\}$ for physics simulation, where $p_0, v_0, F_0^e$ are the initial particle positions, velocities, and elastic deformation gradients, respectively.

To obtain the 3D representation, we first follow PAC-NeRF [45] to separate foreground objects of interest out of the set of initial image frames by running an image matting algorithm (*e.g.*, [38, 39]). We then adopt the standard training pipeline of 3DGS (optionally with the anisotropy regularizer used in [17, 85] to encourage Gaussian kernels to be more evenly shaped to reduce spiky artifacts) to get a dedicated object reconstruction represented by a set of Gaussian kernels $\{\mathcal{K}(i)\}$.

Regarding the initial particle state, a precise description of object boundaries is critical for the simulator to properly handle the interaction between objects. Thus, we explicitly reconstruct a surface mesh for the foreground object via multi-view surface reconstruction [92]. We uniformly sample points inside the mesh volume as the initial particle positions $p_0$. To acquire the initial particle velocities $v_0$, we additionally use first few frames from the set of visual observations $I$ and back-propagate the visual difference loss $\mathcal{L}_v$ through the differentiable renderer $\mathcal{R}$ and simulator $\mathcal{S}$ to optimize $v_0$ while keeping other parameters fixed. The initial elastic deformation gradients $F_0^e$ are set to the identity matrices. In addition, we compute the mass of particles as the product of a constant density $\rho$ and their volume, which is estimated by taking an average of the object volume over particles inside.

**Particle-GS.** Although prior works [17, 85, 95] demonstrated the feasibility of direct adaptation of 3DGS to particle-based physics with an identical mapping between 3D Gaussian kernels and physical particles, we argue that such a kernel-particle relationship is sub-optimal. In particular, the scene reconstruction achieved by 3DGS is primarily optimized for visual appearance, leading to an unbalanced kernel distribution—sparse in texture-less regions and dense in texture-rich areas. Thus, direct simulation based on the original Gaussian kernels may result in unrealistic outcomes. To ameliorate this issue, we propose Particle-GS, which models the hierarchical relationship between Gaussian kernels and simulation particles via a novel particle binding mechanism.

---

**Algorithm 1:** Particle Binding

**Input:** Gaussian centers $\{x(i)\}_{i=1}^{N_K}$,
Gaussian covariance $\{A(i)\}_{i=1}^{N_K}$,
particle positions $\{p_0(j)\}_{j=1}^{N_P}$,
confidence threshold $\tau$

**Output:** Binding matrix $\mathcal{B}$

1  $\mathcal{B} = \text{zeros}(N_K, N_P)$;
2  **for** $i \leftarrow 1$ **to** $N_K$ **do**
3  $\quad$ **for** $j \leftarrow 1$ **to** $N_P$ **do**
   $\quad\quad$ // Difference vector
4  $\quad\quad$ $d_p = p_0(j) - x(i)$;
   $\quad\quad$ // Mahalanobis distance
5  $\quad\quad$ $d_m = d_p^\top A(i)^{-1} d_p$;
   $\quad\quad$ // Check the threshold
6  $\quad\quad$ **if** $d_m \leq \text{chi2}(\tau)$ **then**
7  $\quad\quad\quad$ $\mathcal{B}(i,j) = 1$;
8  $\quad\quad$ **end**
9  $\quad$ **end**
   $\quad$ // Normalize for each row
10 $\quad$ $\mathcal{B}(i,:) = \mathcal{B}(i,:)/(\text{sum}(\mathcal{B}(I,:)))$;
11 **end**

---

**Particle Binding.** Refer to Algorithm 1, we start by binding each Gaussian kernel $\mathcal{K}(i)$ with a set of nearby particles based on the Mahalanobis distance $d_m$ [10, 57] between the kernel and particle position $p_0(j)$. We pre-define a confidence threshold $\tau$ and use the chi-squared test $\text{chi2}(\cdot)$ [24] to check whether $d_m \leq \text{chi2}(\tau)$ for each kernel-particle pair. We record the relationship between Gaussian kernels and simulation particles as a (sparse) binding matrix $\mathcal{B}$, which is used in Stage III to deform Gaussian kernels according to the physical states of multiple internal particles at subsequent time steps (Section 3.3). Our binding mechanism offers two advantages. First, it makes the visual dynamics grounding more robust to the initial visual representations (*i.e.*, 3DGS reconstruction) of the object of interest. Second, since Gaussian kernels generally grow around object surfaces, each kernel carries local object-part information. The proposed binding mechanism acts as a smooth filter to force the particles belonging to the same object part (*i.e.*, Gaussian kernel) to have relatively uniform physical states, thus facilitating realistic simulation.

### 3.2 Physics Simulation

Given the particle state $s_0 = \{p_0, v_0, F_0^e\}$ at the initial time step, we follow the standard MPM practice to perform the time integration scheme $\mathcal{I}$ for physics simulation from $t = 1$ to $t = T$. To correct expert-defined material models, *i.e.*, $\mathcal{M}_0$, to better align with visual observations, we implement NeuMA as $\mathcal{M}_{\theta_i}^i := \mathcal{M}_0^i + \Delta\mathcal{M}_{\theta_i}^i$, where $i \in \{e, p\}$ denotes either the elastic or the plastic

material model, and $\theta_i$ is the associated trainable parameters. We adopt NCLaw's [53] optimized material models as fixed $\mathcal{M}_0$ and leverage LoRA [30] to fulfill the material adaptation process during physics simulation. We embed NeuMA into the differentiable MPM simulator $\mathcal{S}$ and iteratively predict the next physical states $s_{t+1}$ based on the current state $s_t$. Specifically, the simulation process of NeuMA for a single time step can be decomposed into three successive stages:

1. Computing the first Piolar-Kirchhoff stress $P_t$ (equivalently, the Cauchy stress $\sigma$ or the Kirchhoff stress $\tau$) using the elastic material model $\mathcal{M}_{\theta_e}^e$:

$$P_t = \mathcal{M}_{\theta_e}^e(F_t^e), \tag{4}$$

2. Obtaining the updated positions $p_{t+1}$, velocities $v_{t+1}$, and trial elastic deformation gradient $F_{t+1}^{e,\text{trial}}$ of material points via the time integration of MPM (refer to Appendix E.6 for details):

$$p_{t+1}, v_{t+1}, F_{t+1}^{e,\text{trial}} = \mathcal{I}(p_t, v_t, F_t^e, P_t), \tag{5}$$

3. Post-processing the trial elastic deformation gradient by projecting back to the admissible elastic region using the plastic material model $\mathcal{M}_{\theta_p}^p$:

$$F_{t+1}^e = \mathcal{M}_{\theta_p}^p(F_{t+1}^{e,\text{trial}}). \tag{6}$$

### 3.3 Dynamic Scene Rendering

In this subsection, we detail the dynamic scene rendering process achieved by our Particle-GS, which completes our pipeline of dynamics grounding from pixel-level video data. We first augment the static Gaussian kernels to have time-dependent kernel centers and covariance matrices, *i.e.* we use $\mathcal{K}_t(i) = \{x_t(i), \alpha(i), A_t(i), c(i)\}$ to denote the $i$-th Gaussian kernel at $t$-th time step. Following PhysGaussian [85], we assume that the opacity and the spherical harmonic coefficients are invariant over time. To render the predicted frame at $(t+1)$-th time step for supervised training, we deform the Gaussian kernels according to the binding matrix $\mathcal{B}$ obtained at the initial time step and the particle states output by the simulator $\mathcal{S}$. Concretely, we calculate the deformed Gaussian centers $x_{t+1}$ as:

$$\begin{aligned} \Delta x_{t+1} &= \mathcal{B} \times (p_{t+1} - p_t), \\ x_{t+1} &= x_t + \Delta x_{t+1}. \end{aligned} \tag{7}$$

We adopt a similar update scheme for the Gaussian covariance like [85], but additionally consider the binding relationship between kernels and particles:

$$\begin{aligned} \bar{F}_{t+1}^e &= \mathcal{B} \times F_{t+1}^e, \\ A_{t+1} &= \bar{F}_{t+1}^e A_0 (\bar{F}_{t+1}^e)^\top. \end{aligned} \tag{8}$$

After obtaining the updated Gaussian kernels $\{\mathcal{K}_{t+1}(i)\}$, we splat these kernels onto 2D image plane as in Equation (9) to obtain the synthesized image $\hat{I}_{t+1}$.

### 3.4 Training Details

We perform supervised training upon the output of our differentiable renderer $\mathcal{R}$ to optimize NeuMA in an end-to-end manner based on the rendering loss $\mathcal{L}_v$ defined in Equation (2). During the initial state acquisition stage, we actually run the particle binding two times. We obtain the indices of Gaussian kernels without attached particles in the first run and then initialize new physical particles at these kernel centers to ensure each kernel at least binds to one particle. Otherwise, some kernels will remain static across the timeline and lead to inaccurate visual grounding. We then run the final particle binding to record the binding matrix. We optimize the neural material adaptor $\Delta \mathcal{M}_\theta$ using RAdam [49] optimizer with a cosine learning rate scheduler for $1,000$ iterations for each scene. We choose $\tau = 95\%$ and $\rho = 1,000$ for all experiments unless otherwise specified.

## 4 Experiments

### 4.1 Experimental Setup

**Baselines.** Given that NeuMA is a pioneering study aimed at inferring intrinsic dynamics from camera observations alone, making a fair comparison with existing methodologies is inherently challenging. Consequently, we make a best-effort comparison with the combinations of existing related works to evaluate the proposed method from different aspects.

Table 1: **Quantitative comparison in object dynamics grounding in Chamfer distance.**

| Method | BouncyBall | JellyDuck | RubberPawn | ClayCat | SandFish | HoneyBottle | Average |
|---|---|---|---|---|---|---|---|
| PAC-NeRF [45] | 516.30 | 137.73 | 15.47 | 15.38 | 1.71 | 2.21 | 114.80 |
| NCLaw [53] | 56.13 | 6.32 | 3.31 | 2.45 | 2.61 | 2.26 | 12.18 |
| NeuMA | **1.19** | **3.03** | 1.27 | **1.00** | **0.65** | **0.73** | **1.31** |
| NeuMA *w/* P.S. | 1.45 | 3.74 | **1.25** | **1.00** | 0.80 | 0.84 | 1.51 |
| NeuMA *w/o* Bind | 3.34 | 28.42 | 4.62 | 1.26 | 0.97 | 1.01 | 6.60 |
| NeuMA *w/o* $\Delta\mathcal{M}_\theta$ | 1.87 | 3.63 | 1.42 | 1.36 | 1.21 | 1.23 | 1.79 |

- **PAC-NeRF** [45] is the most relevant work to our research, which inverts material parameters (*e.g.*, Young's modulus) of a heuristic simulator, *i.e.*, MPM [34], from multi-view videos.
- **NCLaw** [53] relies on pre-defined particle data to train a general material model without considering adaptation to new kinds of material. We adopt its pre-trained model as the basic material model (*i.e.*, $\mathcal{M}_0$ in Equation (3)) and compare our method with it to show the effect of material adaptation.
- **NCLaw + Particle-GS** (NCLaw+$\mathcal{R}$): Beyond particle-level comparison with NCLaw, we also conduct visual comparisons by integrating pre-trained NCLaw with our renderer, Particle-GS.
- **NeuMA *w/* Particle Supervision** (NeuMA *w/* P.S.): To evaluate the effect of visual supervision, we also report the performance of NeuMA trained with ground-truth 3D particles.
- **NeuMA *w/o* $\Delta\mathcal{M}_\theta$**: We ablate the motion correction term in NeuMA and directly train $\mathcal{M}_0$, in order to verify our core idea — the residual motion adaptation paradigm. This variant is equivalent to *finetuning NCLaw with Particle-GS* using visual observation.
- **NeuMA *w/o* Bind**: We ablate the particle binding procedure by directly treating the 3D Gaussian kernels as physical particles, which is commonly used in previous works [17, 85].
- **Spring-Gaus** [96] is a concurrent work that models elastic objects with Spring-Mass 3D Gaussians. It achieves dynamics grounding by tuning learnable material parameters (*e.g.*, the spring stiffness) with an analytical simulator. We compare it with our method in real-world experiments.

**Dataset.** For a comprehensive evaluation, we consider both synthetic and real-world data.
- **Synthetic Data.** Following PAC-NeRF [45] and NCLaw [53], we use MPM [34] to simulate 6 kinds of dynamics with different initial conditions (object shape, velocities, positions), materials, and time intervals between simulation steps. We use Blender [9] to render high-fidelity and realistic images. The generated 6 benchmarks are named "BouncyBall", "JellyDuck", "RubberPawn"[1], "ClayCat", "HoneyBottle", and "SandFish". We report the simulation details in Appendix B.
- **Real-world Data.** We adopt the real-world data provided by Spring-Gaus [96] to assess the visual grounding performance in the real world. We choose 4 scenes from the released dataset, *i.e.*, "Bun", "Burger", "Dog" and "Pig", for experiments. Please refer to Section 4.1 of [96] for details.

**Metrics.** Following previous works[26, 78], we calculate the L2-Chamfer distance [15, 52] between the grounded and ground-truth particles to measure the accuracy of object dynamics grounding. Note that we scale the values by $10^4$ in all experiments unless otherwise specified. We follow 3DGS [37] and use PSNR [29], SSIM [81], and LPIPS [94] as the metrics for dynamic scene rendering.

**Implementation Details.** In the initial state acquisition stage, we generally obtain 10k~30k Gaussian kernels after static scene reconstruction. We cull Gaussians whose opacities fall behind 0.02 before training NeuMA. The number of initial particles achieved for physics simulation is about 30k. We use 1- and 3-view videos on synthetic and real-world data, respectively, as ground-truth observations for dynamics grounding. In implementing NeuMA, we follow [53] to explicitly regularize neural networks to adhere to two physical priors, *i.e.*, frame indifference and undeformed state equilibrium. Please refer to [53] for the implementation. In addition, we set the rank $r$ and $\alpha$ value of $\Delta\mathcal{M}_\theta$ to 16 by default. Our MPM simulator operates in a $[0, 1]^3$ cube with a fixed resolution of $32^3$ and $70^3$ for synthetic and real-world data unless otherwise specified. All the experiments are conducted on a single NVIDIA A100 GPU.

## 4.2 Performance on Object Dynamics Grounding

Here, we compare NeuMA with considered baselines on visual dynamics grounding using synthetic data. We report the Chamfer distance between the predicted and the ground-truth particle positions in Table 1. It is observed that NeuMA achieves satisfactory results on each scene and the least Chamfer

---

[1]Pawn by Poly by Google [CC-BY] via Poly Pizza.

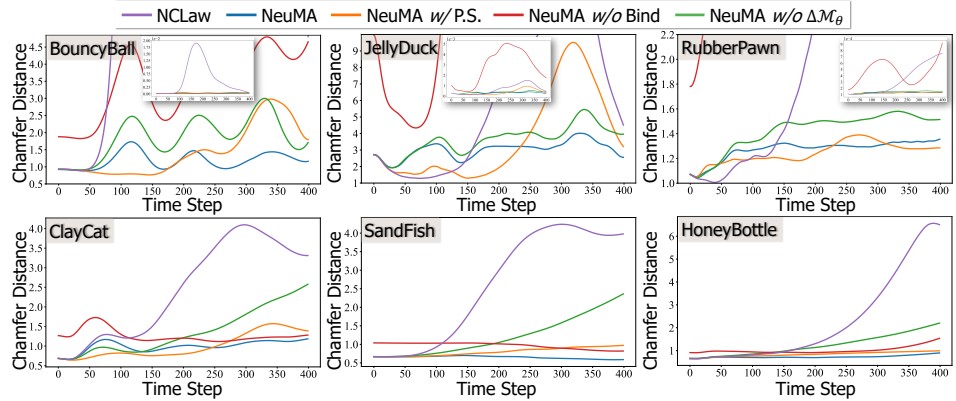

Figure 3: **Comparison in object dynamics grounding over the entire simulation sequence.**

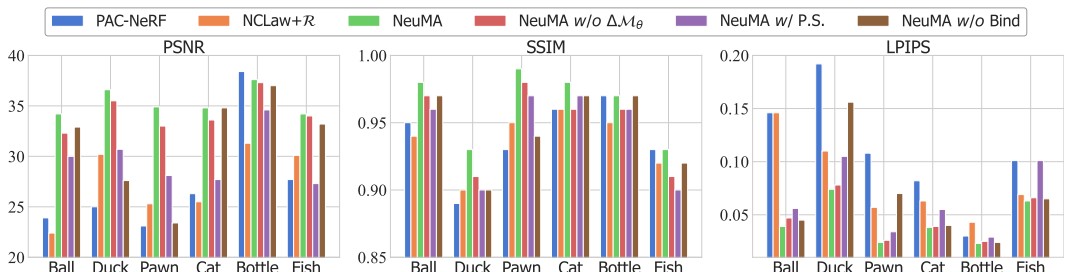

Figure 4: **Quantitative comparison in dynamic scene rendering.**

distance on average, suggesting the superiority of our overall pipeline for visual dynamics grounding. From the first three rows in Table 1, compared to PAC-NeRF [45], which requires an accurate initial guess of the physical parameters, and NCLaw [53], which uses pre-defined fixed material models, NeuMA introduces a learnable residual term $\Delta\mathcal{M}_\theta$ to flexibly correct prior knowledge to align with current observations, significantly improving performance.

We also find that NeuMA even outperforms variants with particle-level supervision (*i.e.*, NeuMA *w/ P.S.*) in some cases. This is a very critical result, proving that the model's reasoning ability can match the human common sense of estimating object dynamics purely from visual observation. We attribute this to the introduction of $\Delta\mathcal{M}_\theta$, which preserves the expert prior of $\mathcal{M}_0$ while fine-tuning to the given dynamic scene. Moreover, results in the fifth row indicate that properly binding physical particles and Gaussian kernels is critical to optimize the residual term $\mathcal{M}_0$. To explain, our particle binding scheme, unlike previous work's one-to-one mapping between particles and Gaussian kernels [85], ensures that particles within a Gaussian kernel share relatively uniform physical states, thereby reducing the degrees of freedom during optimization.

We further show the Chamfer distance at each time step in Figure 3. We observe that the Chamfer distance of NeuMA consistently remains lower than that of other competitors with the simulation ongoing. This verifies that NeuMA can accurately learn the intrinsic dynamics laws of the actual observations. Please refer to the particle visualizations of JellyDuck in Figure 12 for an intuitive comparison. In summary, these results convincingly validate the effectiveness of our main contribution–a learnable, residual neural material adaptor that can correct physical priors from visual observations.

### 4.3 Performance on Dynamic Scene Rendering

In Figure 4, we present averaged visual quality metrics on synthetic data to compare view synthesis performance over the entire simulation time. Our method generally outperforms all baselines, often by a large margin. Furthermore, we present the rendering results in Figure 5. Despite using only single-view video data to ground intrinsic dynamics, our model consistently generates high-fidelity, physically plausible image sequences. In the RubberPawn scene with delicate geometries, PAC-

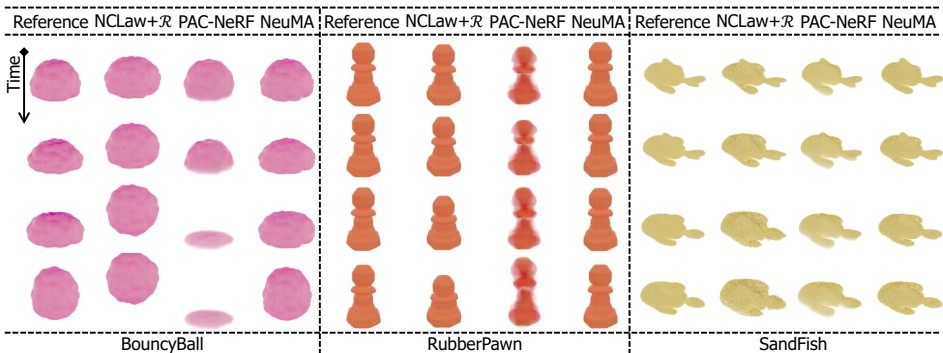

Figure 5: **The visual results for dynamic scene rendering.**

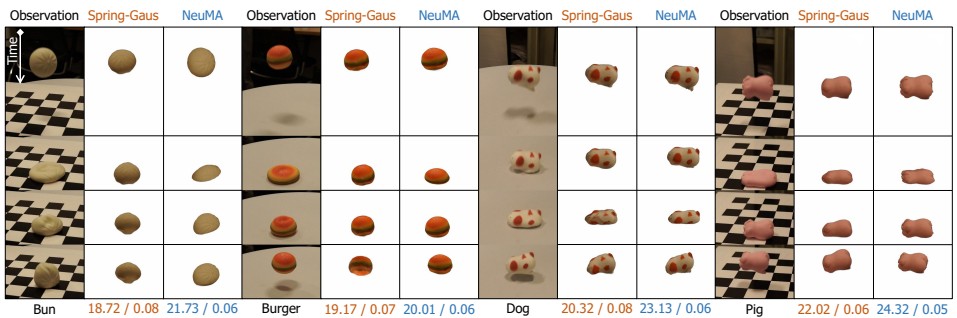

Figure 6: **Comparisons on real-world samples.** We also present quantitative results (*i.e.*, PSNR / LPIPS) between predictions and observations (with background filtered) in the bottom line.

NeRF [45] trained with multi-view data produces blurry artifacts due to limited implicit scene resolution. Although NCLaw [53] combined with our Particle-GS yields better visual quality, it fails to capture material motion accurately. These results verify that our residual adaptation can achieve accurate dynamics grounding and that Particle-GS can handle complex geometries and deformations.

We further evaluate NeuMA on real data in Figure 6. It is seen that the rendered sequences of NeuMA are more aligned with the observations than those of Spring-Gaus [96] both qualitatively and quantitatively. The results confirm the effectiveness of NeuMA in real-world scenarios.

## 4.4 Experimental Analysis

**Dynamics Interpolation.** Recall that $\Delta\mathcal{M}_\theta$ acts as a residual term to correct the expert-designed material model $\mathcal{M}_0$ to match the actual dynamics. To study the flexibility of the residual design, we apply different compositional weights $w = \frac{\alpha}{r}$ [30] to $\Delta\mathcal{M}_\theta$ (by adjusting $\alpha$) and visualize the resulting dynamics in Figure 7. It is observed that NeuMA can smoothly interpolate between the dynamics specified by material priors and by observed images, indicating $\Delta\mathcal{M}_\theta$ indeed learns the difference between the actual dynamics and the dynamics described by the expert-designed motion equations. These visual results verify the flexibility of our proposed residual adaptation paradigm (*i.e.*, $\mathcal{M}_\theta := \mathcal{M}_0 + \Delta\mathcal{M}_\theta$) for grounding intrinsic dynamics.

**Dynamics Generalization.** To study whether NeuMA has learned intrinsic dynamics, we directly apply the trained NeuMA to predict the dynamics given a new object with various initial conditions (*e.g.*, initial velocities and locations). We use the letters of "NeurIPS" as the test target, and the results are shown in Figure 8(a). The used material model is listed at the top. It is observed that new objects also have similar motion patterns as the applied dynamics, verifying that NeuMA indeed learns the intrinsic dynamics that can generalize to novel conditions. Furthermore, we ask: can the learned NeuMA be directly applied to multi-object interaction? The visual results in Figure 8(b), particularly the challenging case of the SandFish colliding with the JellyDuck, show that two trained NeuMA instances can seamlessly interact with each other to produce physically plausible dynamics.

We also quantify the generalization performance in these two scenarios by (1) applying the learned NeuMAs to a new object, *i.e.*, the letter "N", and (2) incorporating two learned NeuMAs for multi-

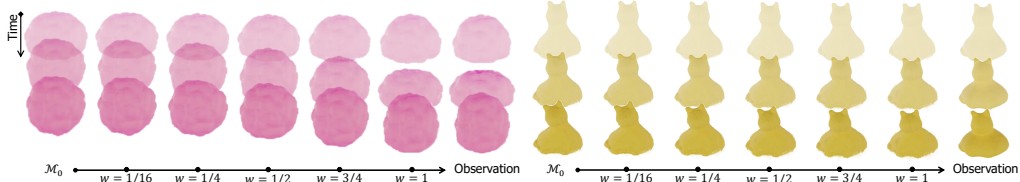

Figure 7: **The visual results for dynamics interpolation.** By applying different weights to the residual term $\Delta \mathcal{M}_\theta$, NeuMA can generate diverse dynamics that smoothly translate prior dynamics to visual observation.

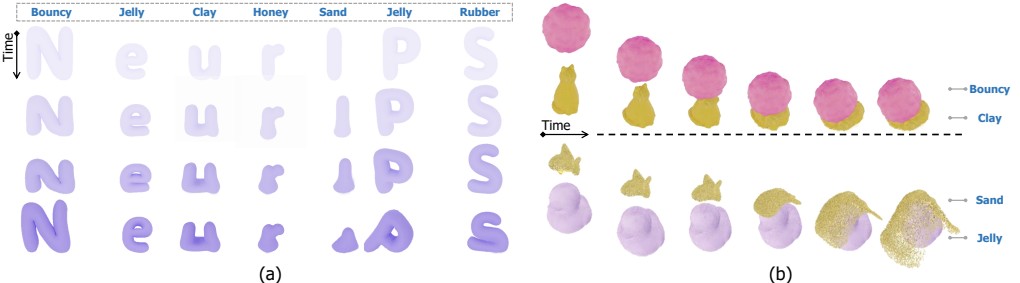

(a)                                                                          (b)

Figure 8: **The visual results for dynamics generalization.** The blue text indicates the applied material for each object. (a) Generalization to new objects (*i.e.* the letters of "NeurIPS"); (b) Generalization to multi-object interactions.

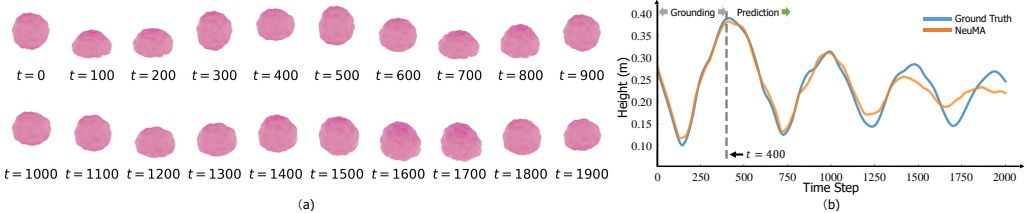

(a)                                                                          (b)

Figure 9: **Results for dynamics prediction.** Given visual data from $t = 0$ to $400$, NeuMA can generate physically plausible prediction for a longer period. (a) Visualization of predicted images; (b) Comparison of the mean height of the BouncyBall between ground truth and our prediction.

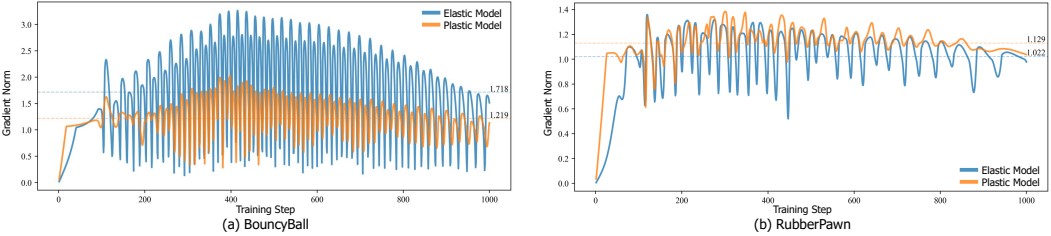

Figure 10: **Gradient norm analysis.** Y-axis: average gradient norms across the training stage. Higher norms indicate more significant changes in the parameters of the corresponding material model.

object interactions. The results in terms of Chamfer distance are shown in Table 2. We observe that NeuMA achieves favorable results over other variants. In summary, the generalization ability of NeuMA largely stems from its refinement of an expert-designed motion model $\mathcal{M}_0$, which ensures physical interpretability and enables effective application in estimating multi-object interactions.

**Dynamics Prediction.** The hallmark of a good physics model is its ability to predict future dynamics accurately [25]. In Figure 9, we investigate NeuMA's predicted dynamics for an extended time. It is seen that NeuMA achieves effective prediction for a period that is three times longer than the given observation sequence, which validates its potential to generate long-duration dynamic videos.

**Prior ($\mathcal{M}_0$) Correction.** Figure 10 presents the gradient norms of the elastic and plastic material models during the training process. As shown in Figure 5, the reference dynamics (*i.e.*, visual

Table 2: **Quantitative results on generalization.**

| Method | Bouncy | Rubber | Sand | Ball & Cat |
|---|---|---|---|---|
| NeuMA | **0.99** | 0.36 | 0.33 | **0.71** |
| NeuMA *w/* P.S. | 1.16 | **0.32** | **0.30** | 0.73 |
| NeuMA *w/o* Bind | 4.26 | 14.99 | 0.48 | 1.19 |
| NeuMA *w/o* LoRA | 1.78 | 0.45 | 0.36 | 0.91 |

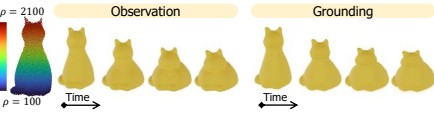

Figure 11: **Grounding result on an object with uneven mass.**

Table 3: **Quantitative results given different physical prior $\mathcal{M}_0$.**

| Setting | $\mathcal{M}_0^e$ | $\mathcal{M}_0^p$ | Grounding | | Generalization | |
|---|---|---|---|---|---|---|
| | | | RubberPawn | ClayCat | Rubber $\rightarrow$ "N" | Clay $\rightarrow$ "N" |
| I | StVK | von Mises | 1.27 | 1.00 | 0.36 | 1.23 |
| II | Neo-Hookean | von Mises | 1.94 | 0.91 | 0.36 | 1.04 |
| III | Fixed Corotated | von Mises | 1.95 | 1.60 | 0.36 | 1.47 |
| IV | Fixed Corotated | Identity | 3.64 | 3.22 | 0.70 | 1.31 |
| V | StVK | Drucker-Prager | 30.26 | 12.91 | 3.91 | 3.31 |

observations) of the BouncyBall deviate from NCLaw (*i.e.*, the prior $\mathcal{M}_0$ in our experiment) in that the reference exhibits more pronounced deformation before returning to its original shape. This deviation is primarily driven by the elastic material model. Interestingly, NeuMA's learning process shows a tendency to adjust the elastic model to better align with the ground-truth observations, as illustrated in Figure 10(a). For the RubberPawn, the main discrepancy from the prior lies in the degree of plastic deformation. As shown in Figure 10(b), NeuMA opts to adjust the plastic model more significantly than the elastic model. In conclusion, this analysis highlights NeuMA's interpretability in adaptively correcting priors on specific material models to better match the visual observations.

**Uneven Mass Distribution.** In our implementation, we, by default, assume a uniform mass distribution for each object. Here, we analyze how NeuMA performs when applied to objects with uneven mass. As shown in Figure 11, we assign particles with different mass densities $\rho$ to obtain a ClayCat with uneven mass. We follow the similar pipeline depicted in Section 3 for visual grounding and display the qualitative results. We also quantify the Chamfer distance between the grounded and ground-truth particles, and the result is $1.33$. From these results, it is seen that NeuMA can handle objects with uneven mass, which validates its potential for visual grounding in complex scenarios.

**Inaccurate Application of** $\mathcal{M}_0$**.** Here, we study the effect of inaccurate application of $\mathcal{M}_0$ using two plastic objects from our synthetic dataset, *i.e.*, RubberPawn and ClayCat. The specific settings and quantitative results (in terms of Chamfer distance) are shown in Table 3. Setting I is our default, where the correct material models are used as the physical prior. Settings II and III introduce inaccurate elastic models, but the resulting motion still follows plastic behavior. Settings IV and V are more challenging, as they are typically used for simulating elastic and granular objects. Our method handles moderate deviations from correct models (Settings II and III), but when the prior is completely incorrect (Settings IV and V), performance drops significantly. To address this, Large Vision Language Models like GPT-4o may be used to infer plausible material models given keyframes.

## 5 Conclusion and Limitations

**Conclusion.** We introduce Neural Material Adaptor (*NeuMA*), a novel framework to infer intrinsic dynamics from visual data. By integrating data-driven corrections with established physical laws, NeuMA combines the interpretability of white box models with the adaptability of black box models. Extensive experiments on object dynamics grounding, dynamic rendering, and its generalizability verify that NeuMA enhances the accuracy and generalization of physical simulations, offering a significant advancement in AI's ability to understand and predict dynamic scenes.

**Limitations.** Despite its strengths, NeuMA has several limitations. First, it requires acquiring initial particles via a multi-view surface reconstruction technique, which, however, requires calibrated cameras and does not perform well in complex scenes. Second, cumulative errors in the forward simulation have a negative effect on future predictions, requiring the design of new physics-informed constraints in the particle space. Additionally, since NeuMA relies on visual supervision, motion blur is a key factor that can degrade the grounding performance.

## Acknowledgments

This work was supported in part by NSFC (62322113, 62376156), Shanghai Municipal Science and Technology Major Project (2021SHZDZX0102), and the Fundamental Research Funds for the Central Universities.

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

## A  Background

**Material Models.** Fundamental to continuum mechanics, material models (also known as constitutive models) define the relationship between material response and the forces or deformations applied. One example is Hooke's law *i.e.*, $F_s = kx$ [68], which relates the spring force with deformation. These models not only encapsulate the material's intrinsic properties but also dictate how it behaves under various loading conditions. In the context of an elastodynamic system following Equation (1), two types of constitutive relationships must be pre-defined. One is the elastic material model $\mathcal{M}^e$ that describes the stress response $\boldsymbol{P}$ given the elastic deformation gradient $\boldsymbol{F}^e$ (*i.e.*, the elastic part of the deformation gradient $\nabla\phi$) [33, 40]. The other is the plastic material model $\mathcal{M}^p$ that defines a return mapping projecting the trial elastic deformation gradient $\boldsymbol{F}^{e,\mathrm{trial}}$ onto the plastic yield constraint $f_Y$ [3, 11]. Traditionally, material models have been constructed through nonlinear polynomial bases [86, 87]. Recently, researchers have adopted learning-based approaches [12, 20, 41, 48, 76] to use neural networks to encode the constitutive relationships for different materials.

**3D Gaussian Splatting.** 3DGS [37] is an optimization-based rasterization approach for time-efficient 3D scene reconstruction. It represents the object of interest as a set of Gaussian kernels $\{\mathcal{K}(i)\}_{i=1}^{N_K}$ with learnable parameters $\mathcal{K}(i) = \{\boldsymbol{x}(i), \alpha(i), \boldsymbol{A}(i), \boldsymbol{c}(i)\}$, where $\boldsymbol{x}(i), \alpha(i), \boldsymbol{A}(i), \boldsymbol{c}(i)$ denote the center, opacity, covariance matrix, and spherical harmonic coefficients of the $i$-th Gaussian kernel. 3DGS splats these kernels onto the 2D image plane to synthesize an image according to the camera viewing matrix. The final color of each pixel is calculated as:

$$C = \sum_i \mathrm{SH}(\boldsymbol{d}(i); \boldsymbol{c}(i)) \cdot \alpha'(i) \prod_{j=1}^{i-1}(1 - \alpha'(j)), \tag{9}$$

where $\alpha'(i) = G(i)\alpha(i)$ is the $z$-depth ordered effective opacity (with $G(i)$ denotes the 2D Gaussian weight of the $i$-th kernel), and $\boldsymbol{d}(i)$ denotes the viewing direction from the camera to $\boldsymbol{x}(i)$. This process is fully differentiable. Hence, 3DGS uses $\ell_1$ loss and structural similarity index measure (SSIM) loss to supervise the optimization of Gaussian kernels.

## B  More Details about the Synthetic Dataset

Table 4: **Typical geometric and physical properties related to the observed dynamics.**

| Benchmark | Initial Shape | Dynamics | Step-size (s) | Initial Velocity (m/s) | Initial Height (m) |
|---|---|---|---|---|---|
| BouncyBall | Ball | Bouncy | $1e-3$ | $(0, -1.92, 0)$ | 0.28 |
| JellyDuck | Duck | Jelly | $1e-3$ | $(0, -1.62, 0)$ | 0.42 |
| RubberPawn | Pawn | Rubber | $5e-4$ | $(0, -1.57, 0)$ | 0.39 |
| ClayCat | Cat | Clay | $5e-4$ | $(0, -2.11, 0)$ | 0.32 |
| HoneyBottle | Bottle | Honey | $5e-4$ | $(0, -1.19, 0)$ | 0.42 |
| SandFish | Fish | Sand | $5e-4$ | $(0, -0.69, 0)$ | 0.28 |

For each benchmark, we use MPM [31, 34, 72] to generate ground-truth particle states for 400 steps and use Blender [9] to render photo-realistic videos at 10 different camera views with a resolution of $800 \times 800$. From these views, we choose one frontal viewpoint of the object for visual grounding on the synthetic data. To ensure an accurate object appearance and geometry, we additionally render 50 uniformly sampled viewpoints at $t = 0$, with the cameras evenly spaced on a sphere covering the object of interest, and use these data for static reconstruction. We also report typical geometric and physical properties of the ground-truth observation in Table 4. Our synthetic dataset features different materials, from elastic objects to granular substances, that exhibit various dynamics and diverse shapes. Please refer to the project page for video demonstrations of the synthetic dataset.

## C  More Experimental Results

**Influence of the Rank of $\Delta\mathcal{M}_\theta$.** Recall that the dynamics corrector $\Delta\mathcal{M}_\theta$ is implemented as a low-rank adaptor (refer to Equation (3)). Here, we evaluate the influence of the rank, *i.e.*, $r$, of $\Delta\mathcal{M}_\theta$ on three synthetic benchmarks. The results are reported in Table 5. Upon reviewing the results, it is

Table 5: **Ablation study on the rank of $\Delta\mathcal{M}_\theta$.** The default rank value is 16.

| Rank of $\Delta\mathcal{M}_\theta$ | BouncyBall | RubberPawn | SandFish | Average |
|---|---|---|---|---|
| 4 | 1.35 | 1.50 | 0.68 | 1.18 |
| 8 | 1.23 | 1.41 | 0.67 | 1.10 |
| 16 | **1.19** | **1.27** | **0.65** | **1.04** |
| 24 | 1.20 | 1.28 | 0.66 | 1.05 |

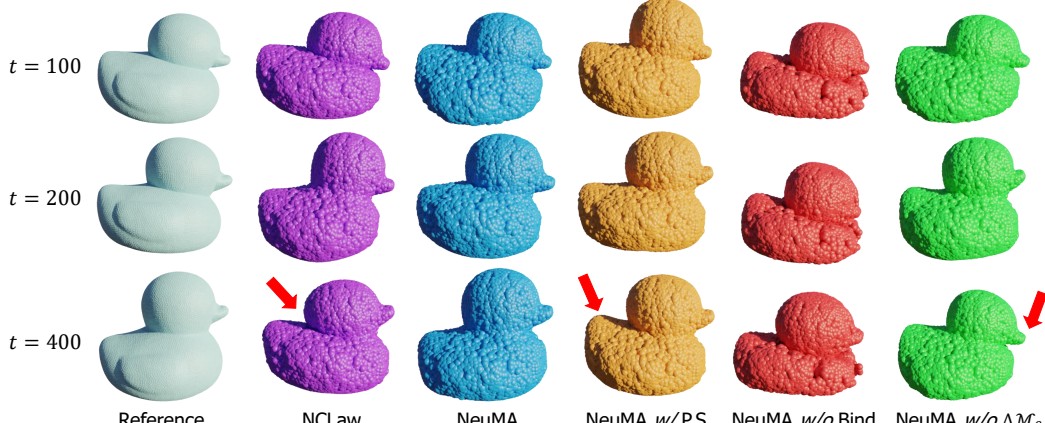

Figure 12: **Visualization of the simulated particles on the JellyDuck benchmark.**

evident that the Chamfer distance increases significantly as the rank decreases. This is likely due to the lower model fitting capacity at lower ranks. However, increasing the rank to a larger value, *e.g.*, 24, does not decrease the chamfer distance further, and even increases slightly. This suggests that the network capacity becomes saturated when the rank exceeds a certain threshold. Therefore, we set the rank as 16 by default in our experiments.

**Visualization of the Physical Particles.** We present the visualization of simulated particles in Figure 12. We choose the JellyDuck benchmark from our synthetic dataset for the following analysis. **(a)** Pre-trained NCLaw fails to generalize to new types of materials that deviate from previously learned priors, as observed by the unrealistic simulation of the motion of the JellyDuck, especially around the duck's neck. This result suggests the necessity of using NeuMA for adaptation to new observations; **(b)** NeuMA *w/* P.S. achieves simulation that is aligned with the reference at several initial steps. However, it suffers from unrealistic simulation afterward, as observed by the tail of the yellow duck at $t = 400$. A possible reason is that simulation at the particle level incurs high freedom during optimization, leading to large cumulative errors as time goes on; **(c)** NeuMA *w/o* Bind exhibits poor results since it directly treats 3D Gaussians as underlying physical particles. However, the distribution of Gaussian kernels may not conform to the continuum hypothesis [8, 23] adopted by particle-based simulators. Thus, driving these kernels directly would result in unrealistic motion; **(d)** Regarding NeuMA *w/o* $\Delta\mathcal{M}_\theta$, direct modification on the parameters of pre-trained material models may overturn the physical priors, as observed by the unrealistic simulation of the green duck's mouth. The result indicates that NeuMA *w/o* $\Delta\mathcal{M}_\theta$ is a sub-optimal way for grounding intrinsic dynamics. This particle-level visualization, from another aspect, demonstrates the superiority of NeuMA in achieving intrinsic dynamics grounding from video data.

**Experiments on complex objects.** Here, we show additional visual results for dynamics grounding on several objects with delicate geometries in Figure 13. Specifically, we consider (a) a large object, "Machine Man"[2], (b) an open container, "Crate", and (c) a non-convex object, "Ring"[3]. While these objects feature different geometric properties, the dynamic rendering results achieved by NeuMA validate its effectiveness on complex scenes.

---

[2]Machine Man by Vaporworks [CC-BY] via Poly Pizza.

[3]Diamond ring by Poly by Google [CC-BY] via Poly Pizza.

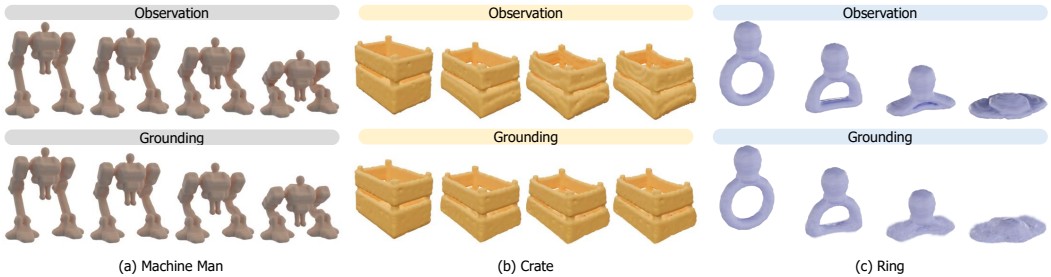

Figure 13: **Visualization of dynamics grounding on complex objects.**

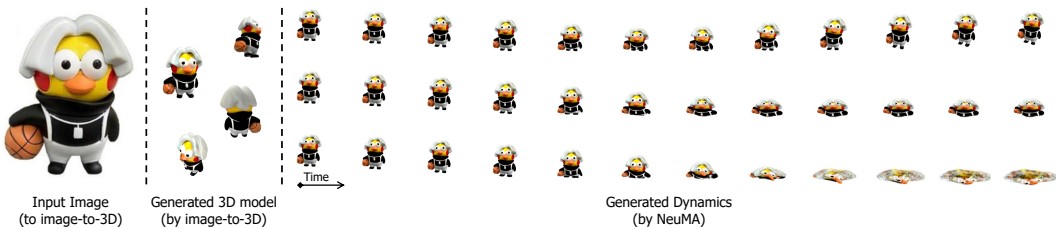

Figure 14: **Visualization of the simulation results on 3D content from an image-to-3D model.**

**Compatibility with Image-to-3D Models.** Recent advancements in 3D content generation [16, 74, 75, 88] offer tremendous potential for various industrial applications, including digital games, virtual reality, and computational design. In this context, an easily conceivable question is whether NeuMA can drive the generated 3D content to produce physically realistic animations. In this subsection, we integrate a state-of-the-art image-to-3D model [88] with NeuMA to investigate such a new pipeline for 4D generation. Concretely, we use a single image as input to the image-to-3D method to obtain a 3D model represented by 3DGS [37] (and optionally do some post-processing to cull the isolated kernels and reduce the number of kernels). We use Blender [9] to transform the 3D representation into a surface mesh for our particle sampling procedure. At this point, we can directly apply trained NeuMAs to generate physically plausible animations, as demonstrated in Section 4.4. We present the visual results of this experiment in Figure 14. From the top to the bottom in the rightmost figure, we separately apply the trained NeuMA on JellyDuck, ClayCat, and HoneyBottle for dynamics generation. As observed, NeuMA is capable of driving the 3D representation produced by the image-to-3D model to generate physically plausible dynamic sequences. These results show NeuMA's potential for physically realistic 4D generation from a single image.

# D  Related Work

**Intuitive Physics.** Intuitive physics studies how humans acquire cognitive capabilities and how to build machine systems with similar capabilities [13, 56]. Researchers explore this topic from various perspectives, including rule-based models [21, 67, 69], probabilistic mental simulation [1, 27], and cognitive intuitive physics engines [2, 5, 77]. With advancements in deep learning and differentiable rendering, intuitive physics has shown promising applicability in reasoning about 3D geometry [55, 61, 82], scene representation [6, 91], physical parameters [7, 28, 46], and object dynamics [19, 43, 84] from images or videos. However, most works focus on rigid objects. Some system identification works [45, 54, 60] are related to ours but are either entangled with geometric shape [54], require extra rendering parameters [60], or rely on expert-defined dynamics models [45]. An open question remains: how can neural networks estimate actual intrinsic dynamics solely from observed images?

**Differentiable Simulation.** Recent works have introduced deep learning-based methods into the physics simulation field, enabling faster and more stable simulations and solving inverse problems. Some works [42, 47, 64, 70, 78] fully embrace neural networks (NN), directly learning an end-to-end neural network to implicitly model physical laws, such as material models [71]. Despite their efficiency, these purely NN-based models cannot guarantee physical correctness and have limited generalizability. Another approach explicitly incorporates expert-designed simulation priors as an

inductive bias in the network architecture [32, 53, 65, 66, 79]. These approaches show great flexibility and generalizability to new shapes and initial/boundary conditions. However, the reliance on known expert knowledge limits their application to arbitrary materials, which yet is the target of our NeuMA.

**Differentiable Rendering.** We modified 3D Gaussian Splatting (3DGS) [37] to serve as our differentiable renderer, enabling image gradient computation relative to simulation results. Unlike other differentiable renderers like mesh-based rasterizers [36, 50, 51], ray tracers [44, 62], and neural radiance fields [4, 18, 58, 59, 63, 80], 3DGS, using explicit 3D Gaussian kernels, is more compatible with physical simulations. Some studies [17, 35, 85, 95, 96] combine 3DGS with physical simulators to generate realistic motions, treating Gaussian kernels as physical particles. However, Gaussian kernels, meant for rendering, can render objects effectively even when unevenly distributed. In contrast, physical simulations require particles to evenly fill objects for accurate representation. Our Particle-GS method calculates the Gaussian kernel and the neighboring relationships of physical particles, adjusting the Gaussian kernel based on nearby physical particles. This decouples property representations, ensuring both accurate rendering and physical simulation.

# E    Introduction to the Material Point Method

## E.1    Introduction

The Material Point Method (MPM) is a numerical technique for solving continuum mechanics problems. Since its first publication in 1994 [72], MPM has greatly matured and become more accurate and stable for simulating the behavior of a wide range of continuum materials. To be self-contained, we provide here an overview of the derivation of MPM from the fundamental partial differential equations (PDEs), which completes the definition of the time integration scheme $\mathcal{I}$ used in Equation (5). For detailed derivations, please refer to previous literature [33, 34, 71, 73, 93].

## E.2    Governing Equations

MPM is derived from the conservation laws of mass and momentum in their Eulerian form:

$$\frac{d\rho}{dt} = -\rho \nabla \cdot \boldsymbol{v}, \tag{10}$$

$$\rho \boldsymbol{a} = \nabla \cdot \boldsymbol{\sigma} + \rho \boldsymbol{b}, \tag{11}$$

where $\rho$ represents density, $\boldsymbol{v}$ is velocity, $\boldsymbol{a}$ denotes acceleration, $\boldsymbol{\sigma}$ is the Cauchy stress tensor, and $\boldsymbol{b}$ stands for body forces. Mass conservation is inherently maintained by advecting the Lagrangian particles in MPM.

## E.3    Weak Formulation

To obtain the weak form of the momentum conservation equation, we multiply it by a test function $\boldsymbol{w}$ and integrate over a domain $\Omega$:

$$\int_{\Omega} \rho \boldsymbol{a} \cdot \boldsymbol{w} d\Omega = \int_{\Omega} (\nabla \cdot \boldsymbol{\sigma}) \cdot \boldsymbol{w} d\Omega + \int_{\Omega} \rho \boldsymbol{b} \cdot \boldsymbol{w} d\Omega. \tag{12}$$

This equation must hold for any test function $\boldsymbol{w}$ and any domain $\Omega$.

Using integration by parts, we convert the weak form into:

$$\int_{\Omega} \rho \boldsymbol{a} \cdot \boldsymbol{w} d\Omega = -\int_{\Omega} \boldsymbol{\sigma} : \nabla \boldsymbol{w} d\Omega + \int_{\partial \Omega_{\boldsymbol{T}}} \boldsymbol{w} \cdot \boldsymbol{T} dS + \int_{\Omega} \rho \boldsymbol{b} \cdot \boldsymbol{w} d\Omega, \tag{13}$$

where $\boldsymbol{T}$ is the traction on the boundary $\partial \Omega_{\boldsymbol{T}}$.

## E.4    Spatial Discretization

MPM discretizes the weak form using basis functions for both the material points and the grid. This results in the following discretized system:

$$\sum_{b=1}^{N_G} M_{ab} \boldsymbol{a}(b) = -\sum_{i=1}^{N_P} V_0(i) \boldsymbol{\tau}(i) \nabla N_a\big(\boldsymbol{x}(i)\big) + \sum_{i=1}^{N_P} M(i) N_a\big(\boldsymbol{x}(i)\big) \boldsymbol{b}(i), \tag{14}$$

---

**Algorithm 2:** Time integration scheme of MPM

---

**Input:** Positions $\boldsymbol{x}_t(i)$, velocities $\boldsymbol{v}_t(i)$, and elastic deformation gradients $\boldsymbol{F}_t^e(i)$ of each material point $i = 1, \ldots, N_P$ at time $t$

**Output:** Positions $\boldsymbol{x}_{t+1}(i)$, velocities $\boldsymbol{v}_{t+1}(i)$, trial elastic deformation gradients $\boldsymbol{F}_{t+1}^{e,\text{trial}}(i)$ of each material point $i = 1, \ldots, N_P$ at time $t+1$

1 **Particle-to-Grid Transfer:** Transfer the Lagrangian particle data to the Eulerian grid by computing for $b = 1, \ldots, N_G$

$$m_t(b) = \sum_{i=1}^{N_P} N_b\big(\boldsymbol{x}_t(i)\big) M(i)$$

$$m_t(b)\boldsymbol{v}_t(b) = \sum_{i=1}^{N_P} N_b\big(\boldsymbol{x}_t(i)\big) M(i)\boldsymbol{v}_t(i)$$

$$\boldsymbol{f}_t^{\boldsymbol{\sigma}}(b) = -\sum_{i=1}^{N_P} \frac{J\big(\boldsymbol{F}_t^e(i)\big)}{\rho_0}\boldsymbol{\sigma}\big(\boldsymbol{F}_t^e(i)\big)\nabla N_b\big(\boldsymbol{x}_t(i)\big) M(i)$$

$$\boldsymbol{f}_t^e(b) = \sum_{i=1}^{N_P} \frac{J\big(\boldsymbol{F}_t^e(i)\big)}{\rho_0}\boldsymbol{b}\big(\boldsymbol{x}_t(i)\big) N_b\big(\boldsymbol{x}_t(i)\big) M(i)$$

2 **Solve Governing Equations on Grid:** Update the grid quantities by computing for $b = 1, \ldots, N_G$

$$\dot{\boldsymbol{v}}_{t+1}(b) = \frac{1}{m_t(b)}\big(\boldsymbol{f}_t^{\boldsymbol{\sigma}}(b) + \boldsymbol{f}_t^e(b)\big)$$

$$\Delta\boldsymbol{v}_{t+1}(b) = \dot{\boldsymbol{v}}_{t+1}(b)\Delta t$$

$$\boldsymbol{v}_{t+1}(b) = \boldsymbol{v}_t(b) + \Delta\boldsymbol{v}_{t+1}(b)$$

3 **Grid-to-Particle Transfer:** Update the particle velocities and deformation gradients by computing for $i = 1, \ldots, N_P$

$$\boldsymbol{v}_{t+1}(i) = \sum_{b=1}^{N_G} N_b\big(\boldsymbol{x}_t(i)\big)\boldsymbol{v}_{t+1}(b)$$

$$\boldsymbol{F}_{t+1}^{e,\text{trial}}(i) = \left(\boldsymbol{I} + \sum_{b=1}^{N_G}\boldsymbol{v}_{t+1}(b) \otimes \nabla N_b\big(\boldsymbol{x}_t(i)\big)\Delta t\right)\boldsymbol{F}_t^e(i)$$

4 **Update Particle Positions:** Update the positions of the particles.

$$\boldsymbol{x}_{t+1}(i) = \boldsymbol{x}_t(i) + \boldsymbol{v}_{t+1}(i)\Delta t$$

---

where $M_{ab} = \sum_{i=1}^{N_P} M(i) \cdot N_a\big(\boldsymbol{x}(i)\big) \cdot N_b\big(\boldsymbol{x}(i)\big)$ is the full mass matrix. Here, $V_0(i)$, $M(i)$, $\boldsymbol{\tau}(i)$, $\boldsymbol{x}(i)$, and $\boldsymbol{b}(i)$ represent the initial volume, mass, Kirchhoff stress, current position, and body force of material point $i$, respectively. $N_b$ and $\boldsymbol{a}(b)$ denote the basis function and acceleration at the Eulerian grid node $b$, with $N_G$ being the number of grid nodes.

## E.5 Temporal Discretization

For time discretization, we use the explicit Euler method with a time step size $\Delta t$:

$$\sum_{b=1}^{N_G} M_{ab}\frac{\boldsymbol{v}_{t+1}(b) - \boldsymbol{v}_t(b)}{\Delta t} = -\sum_{i=1}^{N_P} V_0(i)\boldsymbol{\tau}_t(i)\nabla N_a\big(\boldsymbol{x}_t(i)\big) + \sum_{i=1}^{N_P} M(i)N_a\big(\boldsymbol{x}_t(i)\big)\boldsymbol{b}_t(i). \quad (15)$$

### E.6 MPM Algorithm

We summarize the time integration scheme of the MPM algorithm, implemented under the MLS-MPM framework by [31], in Algorithm 2. This completes the procedure of the physics simulation used in our framework.

