# OpenReview forum: "NeuMA: Neural Material Adaptor for Visual Grounding of Intrinsic Dynamics"
_NeurIPS.cc/2024/Conference — NeurIPS 2024 poster_

### Official Review · Reviewer_Rwta · 2024-07-07

**Soundness:** 2
**Presentation:** 3
**Contribution:** 2
**Rating:** 4
**Confidence:** 5

**Summary:**

This paper introduces a novel pipeline to train the neural constitutive model via image supervisions. It corrects the existing physical material law with a kind of residual learning, and supervise the parameters by aligning groundtruth images the rendered images which are pictured through differentiable renderer techniques.

**Strengths:**

The work conceptually belongs to neural physics dynamics field, which could be somewhat encouraged. It combines the conventional physics knowledge with currently popular neural rendering techniques, might inspire both vision and physics area.

1.	This paper is well-written and is friendly to those who don’t have enough background in material mechanics.

2.	This paper chooses to learn the correction terms via LoRA instead of training the neural constitutive law from scratch, which maintains the original physical priors well and decrease the training difficulties.

3.	The learned material laws can somewhat generalize to large-gap scenes and produce physically plausible motions with various materials.

**Weaknesses:**

Here are still some technical doubts about this method.

1.	The dataset should be described in more detail. For example, the number of views to initialize the scene, and the number of views for the subsequent optimization. How many episodes are used to train each kind of material? These factors are important because they will tell readers the efficiency of the learning-based material law.

2.	In Appendix C, the authors claim that they provide videos in supplementary materials, but I haven’t found them in the submission system.

3.	In Table 1, Fig. 3, and Fig. 4, the results show that the NeuMA overall outperforms the NeuMA w/ PS. I think this could be a bit weird. The groundtruth image sequences are generated by the the groundtruth 3D particle tracks. Thus, the information contained in the particle supervision should not be less than that in the 2D image supervision. However, the results shown by the authors illustrate that only using 2D image supervisions is better than using the 3D particle correspondence. I think the metrics of the model trained by the 3D labels should be the upper limitation of the methods and the result of 2D supervised methods should try to approximate it but cannot entirely surpass it so much. Therefore I think the author did not train the 3D baseline thoroughly, which may be due to the improper selection of training strategy or base model.

4.	The lack of real-world examples could be a small but not severe issue to prove the practicability of the proposed method. If possible, the author should provide some realistic examples. If not, the author should claim the reason for the limitation.

I will reconsider the score according to the answers of the author

**Questions:**

See weakness

**Limitations:**

Yes

---

> ### Author Rebuttal · Authors · 2024-08-07
>
> Thank you for reviewing our work. Your feedback is instrumental in strengthening our paper. Here are our responses to your concerns.
>
> > W1: The dataset should be described in more detail. For example, the number of views to initialize the scene, and the number of views for the subsequent optimization. How many episodes are used to train each kind of material? These factors are important because they will tell readers the efficiency of the learning-based material law.
>
> Thank you for your thoughtful consideration. Each scene from our synthetic data contains 50 uniformly sampled viewpoints for initial state estimation, with the cameras evenly spaced on a sphere covering the object of interest. In addition, we record 400 frames depicting the object’s subsequent motion from a single view. We use the first 20 frames from the dynamic observation to infer the object's initial velocity and the entire trajectories for material adaptor optimization. We train each material adaptor for 1,000 iterations.
>
> > W2: In Appendix C, the authors claim that they provide videos in supplementary materials, but I haven’t found them in the submission system.
>
> We apologize for not managing to upload the video demonstrations of our synthetic dataset at submission. As a remedy, we supplement some key frames to show the object motion in **Figure C in the attached PDF**. We will make sure to release the synthetic data later.
>
> > W3: In Table 1, Fig. 3, and Fig. 4, the results show that the NeuMA overall outperforms the NeuMA w/ PS. I think this could be a bit weird. The groundtruth image sequences are generated by the the groundtruth 3D particle tracks. Thus, the information contained in the particle supervision should not be less than that in the 2D image supervision. However, the results shown by the authors illustrate that only using 2D image supervisions is better than using the 3D particle correspondence. I think the metrics of the model trained by the 3D labels should be the upper limitation of the methods and the result of 2D supervised methods should try to approximate it but cannot entirely surpass it so much. Therefore I think the author did not train the 3D baseline thoroughly, which may be due to the improper selection of training strategy or base model.
>
> Thoughtful viewpoint! The mentioned results may seem weird at first glance but can actually be attributed to the following aspects.
> - **Texture information**: Although the image sequences are generated from ground-truth particles, they are rendered with color under a given light source, which contains information about texture and shading that helps regularize the optimization of the neural material adaptor.
> - **Particle binding**: Note that NeuMA *w/* P.S. does not use the differentiable renderer and, therefore, cannot leverage the particle binding scheme proposed in Section 3.1 of the manuscript. In contrast, our method explicitly models the relationship between 3D Gaussian kernels and simulation particles via the binding mechanism. Since Gaussian kernels generally grow around object surfaces, each kernel encapsulates local object-part information. Our binding mechanism ensures that particles within a kernel share relatively uniform physical states, thereby reducing the degrees of freedom during optimization. NeuMA *w/* P.S., however, exhibits a higher degree of freedom as it is directly supervised by the 3D positions of tens of thousands of particles. This increased complexity can make the optimization of NeuMA *w/* P.S. challenging, especially given that the trainable parameters are of low rank.
>
> Please note that in our previous experiments, we adopt the same hyperparameter settings and the number of training iterations for both NeuMA and NeuMA *w/* P.S. To investigate whether the baseline model is not thoroughly tuned, we conduct additional ablation studies using the BouncyBall benchmark on the learning rate of NeuMA *w/* P.S. in the table below. We report the L2-Chamfer distance (L2CD) between the grounded and ground-truth particles here. These values are scaled by $10^4$.
>
>  ori. lr $\times 0.25$ | ori. lr $\times 0.5$ | ori. lr | ori. lr $\times 1.5$ | ori. lr $\times 2.0$ |
>  :---: | :---: | :---: |:---:|:---:|
>  1.69                  | 1.62                 | 1.45    | 2.00                 | 2.32                 |
>
> From the table, it is observed that the results obtained by these variants still underperform our method (with a value of 1.19).
>
> It is also worth noting that a previous method, PAC-NeRF [44], also observes a similar phenomenon: A 2D pixel-level supervision better optimizes the physical parameters as opposed to a 3D metric. Based on the above analysis, we argue the experimental results should be plausible.
>
> > W4: The lack of real-world examples could be a small but not severe issue to prove the practicability of the proposed method. If possible, the author should provide some realistic examples. If not, the author should claim the reason for the limitation.
>
> Following your suggestions, we supplement dynamics grounding results on real-world data. Specifically, we use the data collected by Spring-Gaus and adopt its experimental setting for experiments. Please kindly refer to **Figure A in the attached PDF** where we show the grounding results achieved by NeuMA and Spring-Gaus. Note that for real-world data, the ground-truth particles are unavailable and thus we do not present the Chamfer distance here. It is observed that the rendered sequences of NeuMA are more aligned with the observations than those of Spring-Gaus both qualitatively and quantitatively. The results confirm the effectiveness of NeuMA in real-world scenarios.

---

> > ### Author Response · Authors · 2024-08-12
> > **Happy to answer any further questions!**
> >
> > Dear Reviewer Rwta,
> >
> > Thank you once again for your insightful comments and suggestions, which helped us improve the quality and clarity of our paper.
> >
> > Following your constructive feedback, we have included more experimental results on real-world data to demonstrate the effectiveness of our method. We have also presented more details about our dataset and explanations of experimental results to provide a better understanding of our proposed method.
> >
> > As the author-reviewer discussion period will end in a few days, we would appreciate it if you could spare some valuable time to have a brief discussion with us. We deeply value and appreciate your feedback and advice.
> >
> > Best,
> >
> > Authors of submission 6610

---

### Official Review · Reviewer_zoZv · 2024-07-11

**Soundness:** 3
**Presentation:** 3
**Contribution:** 3
**Rating:** 7
**Confidence:** 2

**Summary:**

This paper proposes the Neural Material Adaptor (NeuMA), a framework which integrates existing physical laws with learned corrections, thus facilitating the accurate learning of actual dynamics, while also maintaining generalizability and interpretability of physical priors. The framework also proposes Particle-GS, a particle-driven 3D Gaussian Splatting variant, bridging between simulation and observed images, and thus permitting to back-propagate image gradients to optimize the simulator. Various experiments on different dynamics (in terms of grounded particle accuracy, novel-view quality, and generalization ability), demonstrate that NeuMA can accurately capture intrinsic dynamics

**Strengths:**

1. New correction term for the material law.
2. New binding mechanism between particles and Gaussian kernels.
3. Integration with simulation and rendering.
4. Increased accuracy and generalizability.

**Weaknesses:**

1.Acquisition of the initial-state particles requires calibrated cameras not performing well on complex scenes.
2. Reducing cumulative errors in the forward particles simulation.

**Questions:**

Does your framework involve the same number of particles if the objects do not deform in normal situations?

**Limitations:**

1.Acquisition of the initial-state particles requires calibrated cameras not performing well on complex scenes.
2. Reducing cumulative errors in the forward particles simulation.

---

> ### Author Rebuttal · Authors · 2024-08-07
>
> Thank you for your supportive evaluation of our work. We provide the answers to your question below.
>
> > Q1: Does your framework involve the same number of particles if the objects do not deform in normal situations?
>
> Yes. Following the common practice in differentiable physics [25,44,50,88], our method assumes the number of physical particles is invariant across the entire simulation trajectory.

---

> > ### Author Response · Authors · 2024-08-12
> > **Happy to answer any further questions!**
> >
> > Dear Reviewer zoZv,
> >
> > Thank you once again for your insightful comments and suggestions, which helped us improve the quality and clarity of our paper.
> >
> > Following your constructive feedback, we clarified the details of our implementations according to your question.
> >
> > As the author-reviewer discussion period will end in a few days, we would appreciate it if you could spare some valuable time to have a brief discussion with us. We deeply value and appreciate your feedback and advice.
> >
> > Best,
> >
> > Authors of submission 6610

---

### Official Review · Reviewer_TxrR · 2024-07-12

**Soundness:** 3
**Presentation:** 3
**Contribution:** 3
**Rating:** 7
**Confidence:** 4

**Summary:**

The authors propose a new framework to introduce motion dynamics with residuals using a single view on top of a 3D Gaussian splatting based reconstruction of an object obtained from multiple views from the same camera. This approach models motion using physical laws and learned residuals which allows for interpretability while also allowing for transferring material properties to new objects. The authors evaluate their method against existing methods on a variety of materials and geometry and demonstrate a consistent improvement in both dynamics and novel view quality.

**Strengths:**

The paper is mostly well written and easy to follow but a few parts gloss over details relevant to the design decisions.  The experiments are well designed, compared to multiple relevant existing methods and cover both the dynamics and novel view synthesis aspect. Ablations of various aspects of the technique demonstrate their respective importance. The technique goes from modeling motion directly to modeling motion residuals which improves the quality of the reconstruction over time and reduces accumulated errors visible in prior works.

**Weaknesses:**

The method appears to not be evaluated on real-world data where assumptions about motion dynamics may not hold perfectly. It would be interesting to visualize the motion residuals and quantify the errors to better understand their characteristics over time. The paper glosses over some relatively minor design decisions such as how is the single view for grounding dynamics chosen, dimensions of the geometry volume and applying the method to a new object.

**Questions:**

- Uniformly sampling points inside the object to gather the initial state seems to assume that the geometry is a closed volume. What would happen if the given object is not guaranteed to be closed? Have you considered other sampling methods - for example near the surface? How would these other sampling methods impact quality?
- The initial state of Gaussian kernels is constructed from multiple views but the intrinsic dynamics are inferred from a single view. How is this view chosen?
- How would the technique behave when applied to geometry with uneven mass distribution?
- What assumptions are made when applying a pretrained NeuMA to a new object?
- What are some failure cases where the initial state estimate may not produce good results?

**Limitations:**

The authors list a few limitations including the need for calibrated multi-view cameras for the initial 3D reconstruction, the accumulation of errors over time and susceptibility to errors due to motion blur.

The objects used in the simulations appear to be small in size and of uniform material properties. It would be interesting to have some results on larger objects where motion from one end of the object doesn't directly affect the other end or the material varies through the volume.

Another interesting set of evaluations would be on non-convex objects that change topology over time. These would likely highlight limitations of the initial 3D reconstruction which stays constant over time.

---

> ### Author Rebuttal · Authors · 2024-08-07
>
> Thanks for identifying our work and providing valuable comments. We try to address your concerns below.
>
> > W1: The method appears to not be evaluated on real-world data where assumptions about motion dynamics may not hold perfectly. It would be interesting to visualize the motion residuals and quantify the errors to better understand their characteristics over time. The paper glosses over some relatively minor design decisions such as how is the single view for grounding dynamics chosen, dimensions of the geometry volume and applying the method to a new object.
>
> **Real-world data.** We supplement the real-world experiment using data captured by Spring-Gaus and present the result in **Figure A in the attached PDF**. We observe that NeuMA's grounded dynamics are more accurate than the competitor in terms of view synthesis metrics, which verifies the effectiveness of NeuMA in the real world. Note that ground-truth particles are unavailable for real data, so we do not present the Chamfer distance here.
>
> **Visualization of motion residuals.** We show visual results of motion residuals in **Figure D in the attached PDF** to quantify the dynamics grounding errors. The color of each particle indicates its sided Chamfer distance to the ground truth. Please also refer to Figure 3 of the manuscript, where we display the average motion residuals per time step over the simulation trajectory. These results verify the superiority of our method over baselines.
>
> **Single view Choice.** Please refer to `Q2` below.
>
> **Geometry volume.** We use 3D geometry volume, and the grid size is $32^3$.
>
> **Generalization procedure.** The procedure of applying NeuMA to a new object is: (1) Acquire the 3D Gaussian and surface mesh of the new object from multi-view images (details refer to Section 3.1); (2) Set a user-defined initial velocity and select a material adaptor $Δ\mathcal{M}_\theta$ with the physical prior $\mathcal{M}_0$ to generate the physical-plausible animation.
>
> > Q1: Uniformly sampling points inside the object to gather the initial state seems to assume that the geometry is a closed volume. What would happen if the given object is not guaranteed to be closed? Have you considered other sampling methods - for example near the surface? How would these other sampling methods impact quality?
>
> **Open volume.** Thoughtful viewpoint! We use the Material Point Method (MPM) for differentiable simulation in 3D space and require the 3D volume to be set for each particle. Thus, it is difficult for our method to handle curves or thin surfaces without 3D volume. However, if the object has a well-defined 3D volume, our method should work for either closed or open objects. To verify this, we present the dynamics grounding result of an open container in **Figure F in the attached PDF**. It is observed our method can address open volume.
>
> **Particle sampling.** We assume the object is filled with mass, so we use volume sampling instead of surface sampling, since the latter could lead to unrealistic simulations if the interior space is large. For an intuitive understanding, please refer to the results of "NeuMA *w/o* Bind" shown in Figure 10 of the manuscript. In previous experiments, NeuMA *w/o* Bind could be considered as using the surface sampling since we directly treat Gaussian kernels (sprinkled near the object surface) as physical particles for dynamics grounding. For objects with thin structures, like the open container in Figure F, surface and volume sampling yield similar results, and thus their dynamic grounding results are comparable. We also present the L2-Chamfer distance (L2CD) between the grounded and ground-truth particles in this case.
>
> | Open Container | Volume Sampling | Surface Sampling |
> | :---: | :---: | :---: |
> |L2CD ($\times 10^{-4}$)| 1.28 | 1.51 |
>
> > Q2: The initial state of Gaussian kernels is constructed from multiple views but the intrinsic dynamics are inferred from a single view. How is this view chosen?
>
> We use multi-view static observations required by vanilla 3D Gaussian Splatting (3DGS) to accurately capture the object's appearance and geometry. For the single-view dynamic observation, we typically select a frontal view of the object and try to minimize self-occlusion. Note that our method also supports multi-view dynamic observation if such data is available.
>
> > Q3: How would the technique behave when applied to geometry with uneven mass distribution?
>
> We conduct an experiment on an object with uneven mass shown in **Figure E in the attached PDF** by assigning particles with different densities, *i.e.*, $\rho$. We also quantify L2CD in this case, and the result is $1.33\times 10^{-4}$. These results show that NeuMA can handle objects with uneven mass.
>
> > Q4: What assumptions are made when applying a pretrained NeuMA to a new object?
>
> We assume the new object's 3D representations (*i.e.,* 3D Gaussian kernels and surface mesh) are available. Alternatively, we can do 3D reconstruction to get these representations given dense multi-view images. The initial velocity is also required but is user-defined.
>
> > Q5: What are some failure cases where the initial state estimate may not produce good results?
>
> When acquiring the 3D representations, our framework inherits the common failure cases from multi-view reconstruction techniques, *e.g.,* (1) the camera parameters are inaccurate, and (2) the static captures are too sparse. Besides, the prediction of initial velocity may be inaccurate if the object exhibits complex motions at the beginning (*e.g.*, different parts have different initial velocities), as we currently assume a uniform initial velocity.
>
> > Limitation: Large objects and non-convex objects.
>
> Nice suggestion! In **Figure G, H in the attached PDF**, we show the dynamics grounding results of a large object and a non-convex object separately. The quantitative results are $0.91\times 10^{-4}$ and $0.51 \times 10^{-4}$. The results verify the effectiveness of NeuMA on complex geometries.

---

> > ### Author Response · Authors · 2024-08-12
> > **Happy to answer any further questions!**
> >
> > Dear Reviewer TxrR,
> >
> > Thank you once again for your insightful comments and suggestions, which helped us improve the quality and clarity of our paper.
> >
> > Following your constructive feedback, we have included more experimental results on real-world data as well as objects with complex geometries and physical properties. We have also presented more details about the implementations to provide a better understanding of our proposed method.
> >
> > As the author-reviewer discussion period will end in a few days, we would appreciate it if you could spare some valuable time to have a brief discussion with us. We deeply value and appreciate your feedback and advice.
> >
> > Best,
> >
> > Authors of submission 6610

---

> > ### Comment · Reviewer_TxrR · 2024-08-13
> >
> > Thanks for the detailed rebuttal. Since this addresses my questions, I'll update my review.

---

### Official Review · Reviewer_se8S · 2024-07-12

**Soundness:** 3
**Presentation:** 3
**Contribution:** 3
**Rating:** 6
**Confidence:** 3

**Summary:**

NeuMA is a technique to learn residuals on top of physics models to better capture intrinsic dynamics of non-rigid materials.  The paper uses Gaussian splatting (GS) to obtain differentiable rendering, and update the NeuMA model by minimizing reconstruction error of visual observations.  The paper also proposes the use of a particle alignment step that allows simulating more uniformly through the material with minimal impact on the position of the Gaussians.  Evaluations show improved performance over similar approaches and ablations.

**Strengths:**

The paper is well structured, and the methods explained well.  The diagrams are informative and clear.  The results seem to be qualitatively interesting, and quantitatively superior to the baselines considered.  The particle-GS step is a clever method to bind particle-based simulation with Gaussian splatting.

**Weaknesses:**

1. It is a bit unclear to me how the generalization to novel objects and object interaction (in section 4.4) should be evaluated.  This is one of the more exciting uses of the NeuMA approach, but gets very little attention in the paper.  I would appreciate a bit more exploration of how this generalization.  Can these experiments be quantified and compared to other methods?  See the next point for one additional possible ablation.
1. NeuMA interpolates between black- and white-box approaches arbitrarily with a parameter, $\alpha$.  It is unclear what role $\alpha$ plays in the NeuMA approach.  From the provided examples in Figures 8 and 9, it seems that (qualitatively), reconstruction improves as more weight is given to $\Delta\mathcal{M}_\theta$.  This raises the question, what is the benefit of $\mathcal{M}_0$?  Have the authors done experiments with only the learned model?

**Questions:**

1. Emphasis is placed throughout the paper on single-camera inputs, but the initial shape creation requires multiple views.  Is this not contradictory?  I see that distinctions are required because previous approaches require multiple views of the full dynamic trajectory.
1. Does the material need to be known for correct application of $\mathcal{M}_0$?  What is the effect of a misalignment of actual material and the heuristically selected model?
1.  Does the binding between simulated and GS particles need to be updated as timesteps increase and the material deforms substantially?

**Limitations:**

Limitations adequately addressed.

---

> ### Author Rebuttal · Authors · 2024-08-07
>
> We are grateful for your positive and constructive comments, and try to address your concerns below.
>
> > W1: It is a bit unclear to me how the generalization to novel objects and object interaction (in section 4.4) should be evaluated ... Can these experiments be quantified and compared to other methods?
>
> We supplement quantitative results on dynamics generalization by evaluating the L2-Chamfer distance (L2CD) between grounded and ground-truth particles in the simulation space. Note that we scale L2CD by $10^4$ in all experiments (same below).
>
> | Method | Bouncy -> "N" | Rubber -> "N" | Sand -> "N" | Ball & Cat |
> | :--- |:---:|:---:|:---:|:---:|
> | NeuMA |   **0.99**    |     0.36      |    0.33     |  **0.71**  |
> | NeuMA *w/* P.S.  |     1.16      |   **0.32**    |  **0.30**   |    0.73    |
> | NeuMA *w/o* Bind |     4.26      |     14.99     |    0.48     |    1.19    |
> | NeuMA *w/o* LoRA |     1.78      |     0.45      |    0.36     |    0.91    |
>
> In this table,
> -  `Bouncy -> "N"` means that we apply the learned NeuMA on BouncyBall directly to the letter "N" and evaluate the generalization performance (the same with `Rubber -> "N"` and `Sand -> "N"`).
> - `Ball & Cat` is the quantitative result of the interaction between the ball and the cat in Figure 7 in the manuscript.
>
> From the table, we observe that NeuMA achieves favorable performance over other methods in dynamics generalization.
>
> > W2: It is unclear what role $α$ plays in the NeuMA approach. From the provided examples in Figures 8 and 9, it seems that (qualitatively), reconstruction improves as more weight is given to $Δ\mathcal{M}_𝜃$. This raises the question, what is the benefit of $\mathcal{M}_0$? Have the authors done experiments with only the learned model?
>
> We implement the neural material adaptor $Δ\mathcal{M}_\theta$ using the low-rank adaptation (LoRA) and $r,α$ are two hyperparameters for LoRA. Specifically, $r$ is the rank of the trainable weights and $α$, which is commonly set to equal to $r$ during training, can be tuned during inference as a scaling parameter to modify the influence of LoRA on the base model. In our case, $α$ is like a weight coefficient on the adaptor: When $α=0$, it means we do not alter our prior (*i.e.,* the base model $\mathcal{M}_0$); when $α$ gets larger, the generated dynamics will become more similar to the given observation. In general, LoRA could not be evaluated alone without the base model. Please refer to `Q2` below, where we study the benefit of $\mathcal{M}_0$ by changing different $\mathcal{M}_0$ for dynamics grounding.
>
> > Q1: Emphasis is placed throughout the paper on single-camera inputs, but the initial shape creation requires multiple views. Is this not contradictory?
>
> We emphasize the use of a single camera as this eases the burden of camera synchronization for capturing the full dynamic trajectory, which is required by previous works like PAC-NeRF. During the initial state acquisition, we can use a single camera to capture multi-view images to ensure an accurate modeling of the object appearance and geometry for later visual grounding. Moreover, thanks to the physical prior $\mathcal{M}_0$, we have a rough guess of the object motion and could perform dynamics grounding given single-view camera observation. We will clarify our setting in the revision.
>
> > Q2: Does the material need to be known for correct application of $\mathcal{M}_0$? What is the effect of a misalignment of actual material and the heuristically selected model?
>
> In previous experiments, we assume a correct material model is known as the prior $\mathcal{M}_0$ for dynamics grounding. It should be noted that the underlying material parameters (*e.g.,* Young’s modulus and Poisson’s ratio) are unknown. Here, we study the effect of inaccurate application of $\mathcal{M}_0$. We choose two plastic objects, RubberPawn and ClayCat, for this experiment. The specific settings and L2CD results are shown below. We also show the visual grounding results for RubberPawn in **Figure B in the attached PDF**.
>
> | Setting | $\mathcal{M}_0^e$ | $\mathcal{M}_0^p$ | RubberPawn | ClayCat | Rubber -> "N" | Clay -> "N" |
> |:---:|:---:|:---:|:---:|:---:|:---:|:---:|
> |    I    |       StVK        |     von Mises     |    1.27    |  1.00   |     0.36      |    1.23     |
> |   II    |    Neo-Hookean    |     von Mises     |    1.94    |  0.91   |     0.36      |    1.04     |
> |   III   |  Fixed Corotated  |     von Mises     |    1.95    |  1.60   |     0.36      |    1.47     |
> |   IV    |  Fixed Corotated  |     Identity      |    3.64    |  3.22   |     0.70      |    1.31     |
> |    V    |       StVK        |  Drucker-Prager   |   30.26    |  12.91  |     3.91      |    3.31     |
>
> In this table,
> - Setting `I` refers to our previous experimental setting in which the correct material model is set as the physical prior.
> - Settings `II` and `III` adopt inaccurate elastic material models, but the resulting motion still conforming plastic material.
> - Settings `IV` and `V` are more challenging, as the former is commonly used for simulating elastic objects and the latter for granular objects.
>
> From the table, we can see that our method can handle moderate deviation from correct material models (*e.g.*, Setting II and III). However, when the physical prior is completely wrong, (*i.e.*, Setting IV and V), the performance would undergo an obvious decrease. As a remedy, it may be helpful to leverage Large Language Vision Models (LLVM) like GPT-4o to give a plausible guess of the material models given some key frames of the visual observation.
>
> > Q3: Does the binding between simulated and GS particles need to be updated as timesteps increase and the material deforms substantially?
>
> For efficiency, we only compute the binding matrix during the initial stage. This strategy also works well for materials with large deformations (*e.g.*, SandFish from our synthetic data).

---

> > ### Comment · Reviewer_se8S · 2024-08-11
> >
> > Thank you for your response.  You have addressed my major concerns, and I will raise my score

---

> > > ### Author Response · Authors · 2024-08-12
> > >
> > > Dear Reviewer se8S,
> > >
> > > Thank you so much for your acknowledgment! We will incorporate all the contents in the response to our revised manuscript.
> > >
> > > Sincerely,
> > >
> > > Authors of submission 6610

---

### Author Rebuttal · Authors · 2024-08-07

We sincerely thank all reviewers for their time and efforts on reviewing the paper. We are excited to see that reviewers recognized the novelty of our technical contribution (Reviewer TxrR, zoZv, Rwta), acknowledged a better performance achieved by our method over baselines (Reviewer se8S, TxrR, zoZv), and found the paper well-structured and easy-to-follow (Reviewer se8S, TxrR, Rwta).

We also appreciate the reviewers for their constructive comments and concerns. In the attached PDF file, we provide additional visualizations for more details. We summarize the contents in the attached file below.

- **Figure A (to Reviewer TxrR, Rwta)** presents comparisons on dynamics grounding using real-world data captured by Spring-Gaus [a], showing that NeuMA achieves favorable performance on real-world dynamics over the baseline.
- **Figure B (to Reviewer se8S)** illustrates the effect of inaccurate application of $\mathcal{M}_0$, showing that NeuMA can tolerate a wrong material prior to some extent.
- **Figure C (to Reviewer Rwta)** displays our synthetic dataset.
- **Figure D (to Reviewer TxrR)** visualizes the motion residuals achieved by different methods to better demonstrate their grounding performance.
- **Figure E (to Reviewer TxrR)** presents NeuMA's dynamics grounding result on an object with uneven mass distribution.
- **Figure F (to Reviewer TxrR)** presents NeuMA's dynamics grounding result on an open container without closed volume.
- **Figure G (to Reviewer TxrR)** presents NeuMA's dynamics grounding result on a large object where motion from one part does not directly affect another part.
- **Figure H (to Reviewer TxrR)** presents NeuMA's dynamics grounding result on a non-convex object with topology changing over time.

Note: Data used in Figures F, G, and H are from Poly Pizza.

[a] Licheng Zhong, Hong-Xing Yu, Jiajun Wu, and Yunzhu Li. Reconstruction and simulation of elastic objects with spring-mass 3D
gaussians. In ECCV, 2024.

---

### Decision · Program_Chairs · 2024-09-25

**Decision:**

Accept (poster)

**Comment:**

Most reviewers agreed that this paper is impactful and advances the state of the art. I believe the concerns of the one hold-out were adequately addressed during the rebuttal period, although they did not respond. For these reasons I recommend acceptance.